

# Model independent feature attributions: Shapley values that uncover non-linear dependencies

Daniel Vidali Fryer[1], Inga Strumke[2] and Hien Nguyen[3]

[1] School of Mathematics Physics, University of Queensland, Queensland, St Lucia, Australia
[2] Department of Holistic Systems, Simula Research Laboratory, Oslo, Norway
[3] Department of Mathematics and Statistics, La Trobe University, Melbourne, Australia

## ABSTRACT

Shapley values have become increasingly popular in the machine learning literature, thanks to their attractive axiomatisation, flexibility, and uniqueness in satisfying certain notions of 'fairness'. The flexibility arises from the myriad potential forms of the Shapley value game formulation. Amongst the consequences of this flexibility is that there are now many types of Shapley values being discussed, with such variety being a source of potential misunderstanding. To the best of our knowledge, all existing game formulations in the machine learning and statistics literature fall into a category, which we name the model-dependent category of game formulations. In this work, we consider an alternative and novel formulation which leads to the first instance of what we call model-independent Shapley values. These Shapley values use a measure of non-linear dependence as the characteristic function. The strength of these Shapley values is in their ability to uncover and attribute non-linear dependencies amongst features. We introduce and demonstrate the use of the energy distance correlations, affine-invariant distance correlation, and Hilbert–Schmidt independence criterion as Shapley value characteristic functions. In particular, we demonstrate their potential value for exploratory data analysis and model diagnostics. We conclude with an interesting expository application to a medical survey data set.

## INTRODUCTION

There are many different meanings of the term "feature importance", even in the context of Shapley values. Indeed, the meaning of a Shapley value depends on the underlying game formulation, referred to by *Merrick & Taly (2019)* as the *explanation game*. Although, this is so far rarely discussed explicitly in the existing literature. In general, Shapley value explanation games can be distinguished as either belonging to the model-dependent category or the model-independent category. The latter category is distinguished by an absence of assumptions regarding the data generating process (DGP). Here, the term model-dependent refers to when the Shapley value depends on a choice of fitted model (such as the output of a machine learning algorithm), or on a set of fitted models (such as the set of sub-models of a linear model).

Corresponding author
Inga Strumke, inga@simula.no

Shapley values that uncover non-linear dependencies (Sunnies) is, to the best of our knowledge, the only Shapley-based feature importance method that falls into the model-independent category. In this category, feature importance scores attempt to determine what is *a priori* important, in the sense of understanding the partial dependence structures within the joint distribution describing the DGP. We show that these methods that generate model-independent feature importance scores can appropriately be used as model diagnostic procedures, as well as procedures for exploratory data analysis.

Existing methods in the model-dependent category, on the other hand, seek to uncover what is *perceived* as important by the model (or class of models), either with regards to a performance measure (e.g., a goodness-of-fit measure) or for measuring local influences on model predictions. Model-dependent definitions of feature importance scores can be distinguished further according as to whether they depend on a fitted (i.e., trained) model or on an unfitted class of models. We refer to these as within-model scores and between-model scores, respectively. This distinction is important, since the objectives are markedly different.

Within-model Shapley values seek to describe how the model reacts to a variety of inputs, while, e.g., accounting for correlated features in the training data by systematically setting "absent" features to a reference input value, such as a conditional expectation. There are many use cases for within-model Shapley values, such as providing transparency to model predictions, e.g. for explaining a specific credit decision or detecting algorithmic discrimination (*Datta, Sen & Zick, 2016*), as well as understanding model structure, measuring interaction effects and detecting concept drift (*Lundberg et al., 2020*).

All within-model Shapley values that we are aware of fall into the class of *single reference games,* described by *Merrick & Taly (2019)*. These include SAGE (*Covert, Lundberg & Lee, 2020*); SHAP (*Lundberg & Lee, 2017*); Shapley Sampling Values (*Štrumbelj & Kononenko, 2013*); Quantitative Input Influence (*Datta, Sen & Zick, 2016*); Interactions-based Method for Explanation(IME) (*Štrumbelj, Kononenko & Šikonja, 2009*); and TreeExplainer (*Lundberg et al., 2020*). Note that some within-model feature importance methods, such as SHAP, can be described as *model agnostic* methods, since they may be applied to any trained model. Regardless, such values are dependent on a prior choice of fitted model.

In contrast to within-model Shapley values, between-model Shapley values seek to determine which features influence an outcome of the model fitting procedure, itself, by repeatedly refitting the model to compute each marginal contribution. Such scores have been applied, for example, as a means for feature importance ranking in regression models. These include Shapley Regression Values (*Lipovetsky & Conklin, 2001*), ANOVA Shapley values (*Owen & Prieur, 2017*), and our prior work (*Fryer, Strumke & Nguyen, 2020*). The existing between-model feature importance scores are all *global* feature importance scores, since they return a single Shapley value for each feature, over the entire data set. Sunnies is also a global score, though not a between-model score.

A number of publications and associated software have been produced recently to efficiently estimate or calculate SHAP values. Tree SHAP, Kernel SHAP, Shapley Sampling Values, Max Shap, Deep Shap, Linear-SHAP and Low-Order-SHAP are all methods

for either approximating or calculating SHAP values. However, these efficient model-dependent methods for calculating or approximating SHAP values are developed for local within-model scores, and are not suitable for Sunnies, which is a global and model-independent score. While Sunnies does not fit under the model-dependent frameworks for efficient estimation, Shapley values in general can be approximated via a consistent Monte Carlo algorithm introduced by *Song, Nelson & Staum (2016)*. While efficient approximations do exist, computational details are not the focus of this paper, where we focus on the concept and relevance of Sunnies.

In "Shapley Decomposition", we introduce the concept of the Shapley value and its decomposition. We then introduce the notion of *attributed dependence on labels* (ADL), and briefly demonstrate the behaviour of the $R^2$ characteristic function on a data set with non-linear dependence, to motivate our alternative measures of non-linear dependence in place of $R^2$. In "Measures of non-linear dependence", we describe three such measures: the Hilbert Schmidt Independence Criterion (HSIC), the Distance Correlation (DC) and the Affine-Invariant Distance Correlation (AIDC). We use these as characteristic functions throughout the remainder of the work, although we focus primarily on the DC.

The DC, HSIC and AIDC do not constitute an exhaustive list of the available measures of non-linear dependence. We do not provide here a comparison of their strengths and weaknesses. Instead, our objective is to propose and demonstrate a variety of use cases for the general technique of computing Shapley values for model-independent measures of statistical dependence.

In "Exploration", we demonstrate the value of ADL for exploratory data analysis, using a simulated DGP that exhibits mutual dependence without pairwise dependence. We also leverage this example to compare ADL to popular pairwise and model-dependent measures of dependence, highlighting a drawback of the pairwise methods, and of the popular XGBoost built-in "feature importance" score. We also show that SHAP performs favourably here. In "Diagnostics", we introduce the concepts of *attributed dependence on predictions* (ADP) and *attributed dependence on residuals* (ADR). Using simulated DGPs, we demonstrate the potential for ADL, ADP and ADR to uncover and diagnose model misspecification and concept drift. For the concept drift demonstration ("Demonstration with concept drift"), we see that ADL provides comparable results to SAGE and SHAP, but without the need for a fitted model. Conclusions are drawn in Section "Discussion and Future Work".

## SHAPLEY DECOMPOSITION

In approaching the question: "How do the different features $X = (X_1, \ldots, X_d)$ in this data set affect the outcome $Y$?", the concept of a Shapley value is useful. The Shapley value has a long history in the theory of cooperative games, since its introduction in *Shapley (1953)*, attracting the attention of various Nobel prize-winning economists (cf. *Roth, 1988*), and enjoying a recent surge of interest in the statistics and machine learning literature. *Shapley (1953)* formulated the Shapley value as the unique game theoretic solution concept, which satisfies a set of four simple and apparently desirable axioms: *efficiency*, *additivity*,

*symmetry* and the *null player* axiom. For a recent monograph, defining these four axioms and introducing solution concepts in cooperative games, consult *Algaba, Fragnelli & Sánchez-Soriano (2019)*.

As argued by *Lipovetsky & Conklin (2001)*; *Israeli (2007)*; *Huettner & Sunder (2012)*, we can think of the outcome $C(S)$ of a prediction or regression task as the outcome of a cooperative game, in which the set $S = \{X_1, \ldots, X_d\}$ of data features represent a coalition of players in the game. The function $C$ is known as the *characteristic function* of the game. It maps elements $S$, in the power set $2^{[d]}$ of players, to a set of payoffs (or outcomes) and thus fully describes the game. Let $d$ be the number of players. The *marginal contribution* of a player $v \in S$ to a team $S$ is defined as $C(S \cup \{v\}) - C(S)$. The average marginal contribution of player $v$, over the set $\mathscr{S}_k$ of all teams of size $k$ that exclude $v$, is

$$\overline{C}_k(v) = \frac{1}{|\mathscr{S}_k|} \sum_{S \in \mathscr{S}_k} [C(S \cup \{v\}) - C(S)], \tag{1}$$

where $|\mathscr{S}_k| = \binom{d-1}{k}$. The Shapley value of player $v$, then, is given by

$$\phi_v(C) = \frac{1}{d} \sum_{k=0}^{d-1} \overline{C}_k(v), \tag{2}$$

i.e., $\phi_v(C)$ is the average of $\overline{C}_k(v)$ over all team sizes $k$.

## Attributed dependence on labels

The characteristic function $C(S)$ in Eq. (1) produces a single payoff for the features with indices in $S$. In the context of statistical modelling, the characteristic function will depend on $Y$ and $X$. To express this we introduce the notation $X|_S = (X_j)_{j \in S}$ as the projection of the feature vector onto the coordinates specified by $S$, and we write the characteristic function $C_Y(S)$ with subscript $Y$ to clarify its dependence on $Y$ as well as $X$ (via $S$). Now, we can define a new characteristic function $R_Y$ in terms of the popular coefficient of multiple correlation $R^2$, as

$$R_Y(S) = R^2(Y, X|_S) = 1 - \frac{|Cor(Y, X|_S)|}{|Cor(X|_S)|}, \tag{3}$$

where $|\cdot|$ and $Cor(\cdot)$ are the determinant operator and correlation matrix, respectively (cf. *Fryer, Strumke & Nguyen, 2020*).

The set of Shapley values of all features in $X$, using characteristic function $C$, is known as the Shapley decomposition of $C$ amongst the features in $X$. For example, the Shapley decomposition of $R_Y$, from Eq. (3), is the set $\{\phi_v(R_Y) : v \in [d]\}$, calculated via Eq. (2).

In practice, the joint distribution of $(Y, X^T)$ is unknown, so the Shapley decomposition of $C$ is estimated via substitution of an empirical characteristic function $\hat{C}$ in Eq. (1). In this context, we work with an $n \times |S|$ data matrix $\mathbf{X}|_S$, whose $i$th row is the vector $\mathbf{x}|_S = (x_{ij})_{j \in S}$, representing a single observation from $X|_S$. As a function of this observed data, along with the vector of observed labels $\mathbf{y} = (y_i)_{i \in [n]}$, the empirical characteristic function $\hat{C}_{\mathbf{y}}$ produces an estimate of $C_Y$ that, with Eq. (1), gives the estimate $\phi_v(\hat{C}_{\mathbf{y}})$, which we refer to as the Attributed Dependence on Labels (ADL) for feature $v$.

### *Recognising dependence: Example 1*

For example, the empirical $R^2$ characteristic function $\hat{R}_{\mathbf{y}}$ is given by

$$\hat{R}_{\mathbf{y}}(S) = 1 - \frac{|\rho(\mathbf{y}, \mathbf{X}|_S)|}{|\rho(\mathbf{X}|_S)|}, \tag{4}$$

where $\rho$ is the empirical Pearson correlation matrix.

Regardless of whether we use a population measure or an estimate, the $R^2$ measures only the *linear* relationship between the response (i.e., labels) $Y$ and features $X$. This implies the $R^2$ may perform poorly as a measure of dependence in the presence of non-linearity. The following example from a non-linear DGP demonstrates this point.

Suppose the features $X_j, j \in [d]$ are independently uniformly distributed on $[-1, 1]$. Given a diagonal matrix $A = \text{diag}(a_1, \ldots, a_d)$, let the response variable $Y$ be determined by the quadratic form

$$Y = X^T A X = a_1 X_1^2 + \ldots + a_d X_d^2. \tag{5}$$

Then, the covariance $\text{Cov}(Y, X_j) = 0$ for all $j \in [d]$. This is because

$$\text{Cov}(X^T A X, X_j) = \sum_{j=1}^{d} \text{Cov}(X_j^2, X_j) = 0,$$

since $\text{E}[X_j] = 0$ and $\text{E}[X_j^3] = 0$. In Fig. 1, we display the $X_4$ cross section of $10,000$ observations generated from Eq. (5) with $d = 5$ and $A = \text{diag}(0, 2, 4, 6, 8)$, along with the least squares line of best fit and associated $R^2$ value. We visualize the results for the corresponding Shapley decomposition in Fig. 2. As expected, we see that the $R^2$ is not able to capture the non-linear dependence structure of Eq. (5), and thus neither is its Shapley decomposition.

We note that improvements on the results in Figs. 1 and 2 can be obtained by choosing a suitable linearising transformation of the features or response prior to calculating $R^2$, but such a transformation is not known to be discernible from data in general, except in the simplest cases.

## Measures of non-linear dependence

In the following, we describe three measures of non-linear dependence that, when used as a characteristic function $C$, have the following properties.

- Independence is detectable (in theory), i.e., if $C(S) = 0$, then the variables $Y$ and $X|_S$ are independent. Equivalently, dependence is visible, i.e., if $Y$ and $X|_S$ are dependent, then $C(S) \neq 0$.
- $C$ is model-independent. Thus, no assumptions are made about the DGP and no associated feature engineering or transformation of $X$ or $Y$ is necessary. Note that, since the Shapley values sum, by efficiency, to the distance correlation, we do get the guarantee that dependence on $[d]$ is visible in the sum of Shapley values, by virtue of equalling $C([d])$. However, each individual Shapley value is not a distance correlation, but a linear combination of distance correlations, and thus cannot itself generally be interpreted as a distance correlation. The same is true for any sum of a strict subset of the Shapley values, since efficiency applies to the sum of *all* Shapley values, and not a strict subset of them.

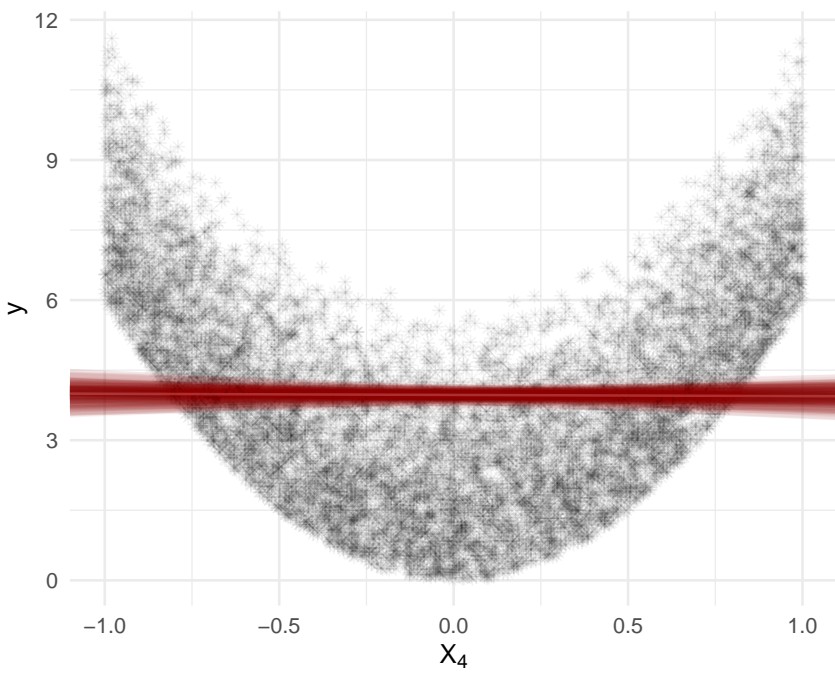

**Figure 1** **Feature $X_4$, from Eq. (5), cross section with 100 least squares lines of best fit, each produced from a random sample of size 1,000, from the simulated population of size 10,000.** The estimate of $R^2$ is 0.0043, with 95% bootstrap confidence interval (0.001, 0.013) over 100 fits. The $R^2$ is close to 0 despite the presence of strong (non-linear) dependence.

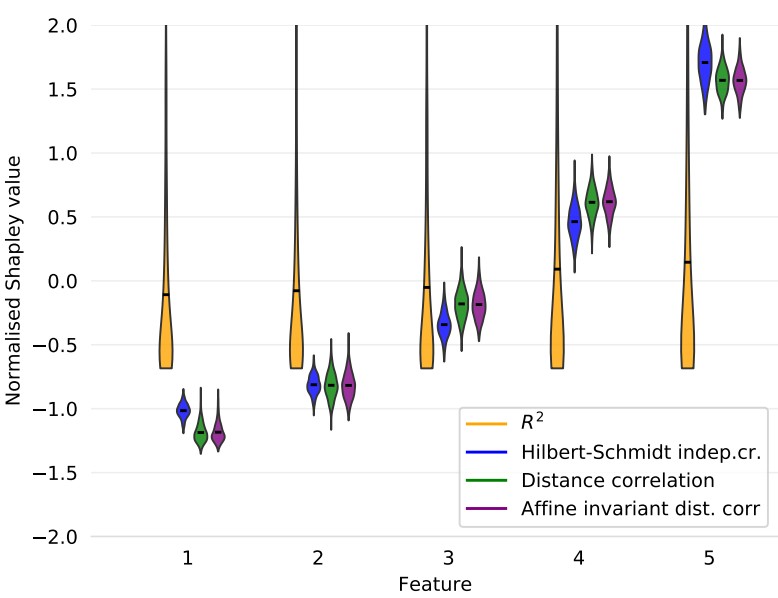

**Figure 2** **Shapley decompositions using the four measures of dependence described in "Measures of non-linear dependence", normalised for comparability, with sample size 1,000 over 1,000 iterations.**

[1] In this context, we refer to the *characteristic function* of a probability distribution. We would like to make the reader aware that this is a different use of the term "characteristic function" than that used to describe a cooperative game in the context of Shapley values, as in Eq. (1).

### Distance correlation and affine invariant distance correlation

The distance correlation, and its affine invariant adaptation, were both introduced by *Székely, Rizzo & Bakirov (2007)*. Unlike the Pearson correlation, the distance correlation between $Y$ and $X$ is zero if and only if $Y$ and $X$ are statistically independent. However, the distance correlation is equal to 1 *only* if the dimensions of the linear spaces spanned by $Y$ and $X$ are equal, almost surely, and $Y$ is a linear function of $X$.

First, the population distance covariance between the response $Y$ and feature vector $X$ is defined as a weighted $L_2$ norm of the difference between the joint characteristic function[1] , $f_{YX}$ and the product of marginal characteristic functions $f_Y f_X$. In essence, this is a measure of squared deviation from the assumption of independence, i.e., the hypothesis that $f_{YX} = f_Y f_X$.

The empirical distance covariance $\mathcal{V}_n^2$ is based on Euclidean distances between sample elements, and can be computed from data matrices $\mathbf{Y}, \mathbf{X}$ as

$$\hat{\mathcal{V}}^2(\mathbf{Y}, \mathbf{X}) = \sum_{i,j=1}^{n} A(\mathbf{Y})_{ij} A(\mathbf{X})_{ij}, \tag{6}$$

where the matrix function $A(\mathbf{W})$ for $\mathbf{W} \in \{\mathbf{Y}, \mathbf{X}\}$ is given by

$$A(\mathbf{W})_{ij} = B(\mathbf{W})_{ij} - \frac{1}{n}\sum_{i=1}^{n} B(\mathbf{W})_{ij} - \frac{1}{n}\sum_{j=1}^{n} B(\mathbf{W})_{ij} + \frac{1}{n^2}\sum_{i,j=1}^{n} B(\mathbf{W})_{ij},$$

where $|| \cdot ||$ denotes the Euclidean norm, and $B(\mathbf{W})$ is the $n \times n$ distance matrix with $B(\mathbf{W})_{ij} = ||\mathbf{w}_i - \mathbf{w}_j||$, where $\mathbf{w}_i$ denotes the $i$th observation (row) of $\mathbf{W}$. Here, $\mathbf{Y}$ is in general a matrix of observations, with potentially multiple features. Notice the difference between $\mathbf{Y}$ and $\mathbf{y}$, where the latter is the (single column) label vector introduced in "Attributed dependence on labels".

The empirical distance correlation $\hat{\mathcal{R}}$ is given by

$$\hat{\mathcal{R}}^2(\mathbf{Y}, \mathbf{X}) = \frac{\hat{\mathcal{V}}^2(\mathbf{Y}, \mathbf{X})}{\sqrt{\hat{\mathcal{V}}^2(\mathbf{Y}, \mathbf{Y})\hat{\mathcal{V}}^2(\mathbf{X}, \mathbf{X})}}, \tag{7}$$

for $\hat{\mathcal{V}}^2(\mathbf{Y}, \mathbf{Y})\hat{\mathcal{V}}^2(\mathbf{X}, \mathbf{X}) \neq 0$, and $\hat{\mathcal{R}}(\mathbf{Y}, \mathbf{X}) = 0$ otherwise. For our purposes, we define the distance correlation characteristic function estimator

$$\hat{D}_{\mathbf{y}}(S) = \hat{\mathcal{R}}^2(\mathbf{y}, \mathbf{X}|_S). \tag{8}$$

A transformation of the form $x \mapsto Ax + b$ for a matrix $A$ and vector $b$ is called affine. Affine invariance of the distance correlation is desirable, particularly in the context of hypothesis testing, since statistical independence is preserved under the group of affine transformations. When $\mathbf{Y}$ and $\mathbf{X}$ are first scaled as $\mathbf{Y}' = \mathbf{Y}S_{\mathbf{Y}}^{-1/2}$ and $\mathbf{X}' = \mathbf{X}S_{\mathbf{X}}^{-1/2}$, the distance correlation $\hat{\mathcal{V}}(\mathbf{Y}', \mathbf{X}')$, becomes invariant under any affine transformation of $\mathbf{Y}$ and $\mathbf{X}$ (*Székely, Rizzo & Bakirov, 2007*, Section 3.2). Thus, the empirical affine invariant distance correlation is defined by

$$\hat{\mathcal{R}}'(\mathbf{Y}, \mathbf{X}) = \hat{\mathcal{R}}(\mathbf{Y}S_{\mathbf{Y}}^{-1/2}, \mathbf{X}S_{\mathbf{X}}^{-1/2}), \tag{9}$$

and we define the associated characteristic function estimator $\hat{D}'_{\mathbf{y}}$ in the same manner as Eq. (8). Monte Carlo studies regarding the properties of these measures are given by *Székely, Rizzo & Bakirov (2007)*.

### Hilbert–Schmidt independence criterion

The Hilbert Schmidt Independence Criterion (HSIC) is a kernel-based independence criterion, first introduced by *Gretton et al. (2005a)*. Kernel-based independence detection methods have been adopted in a wide range of areas, such as independent component analysis (*Gretton et al., 2007*). The link between energy distance-based measures, such as the distance correlation, and kernel-based measures, such as the HSIC, was established by *Sejdinovic et al. (2013)*. There, it is shown that the HSIC is a certain formal extension of the distance correlation.

The HSIC makes use of the cross-covariance operator, $C_{YX}$, between random vectors $Y$ and $X$, which generalises the notion of a covariance. The response $Y$ and feature vector $X$ are each mapped to functions in a Reproducing Kernel Hilbert Spaces (RKHS), and the HSIC is defined as the Hilbert–Schmidt (HS) norm $||C_{YX}||_{HS}^2$ of the cross-covariance operator between these two spaces (*Gretton et al., 2005b*; *Gretton et al., 2007*; *Gretton et al., 2005a*). Given two kernels $\ell, k$, associated to the RKHS of $Y$ and $X$, respectively, and their empirical evaluation matrices $\mathbf{L}, \mathbf{K}$ with row $i$ and column $j$ elements $\ell_{ij} = \ell(y_i, y_j)$ and $k_{ij} = k(\mathbf{x}_i, \mathbf{x}_j)$, where $\mathbf{y}_i, \mathbf{x}_i$ denote the $i$th observation (row) in data matrices $\mathbf{X}$ and $\mathbf{Y}$, respectively, the empirical HSIC can be calculated as

$$\widehat{\text{HSIC}}(\mathbf{Y}, \mathbf{X}) = \frac{1}{n^2} \sum_{i,j}^{n} k_{ij} \ell_{ij} + \frac{1}{n^4} \sum_{i,j,q,r}^{n} k_{ij} \ell_{qr} - \frac{2}{n^3} \sum_{i,j,q}^{n} k_{ij} \ell_{iq}. \tag{10}$$

As in "Distance correlation and affine invariant distance correlation", notice the difference between $\mathbf{Y}$ and $\mathbf{y}$, where the latter is the (single column) label vector introduced in "Attributed dependence on labels". Intuitively, this approach endows the cross-covariance operator with the ability to detect non-linear dependence, and the HS norm measures the combined magnitude of the resulting dependence. For a thorough discussion of positive definite Kernels, with a machine learning emphasis, see the work of *Hein & Bousquet (2004)*.

Calculating the HSIC requires selecting a kernel. The Gaussian kernel is a popular choice that has been subjected to extensive testing in comparison to other kernel methods [see, e.g., *Gretton et al., 2005a*]. For our purposes, we define the empirical HSIC characteristic function by

$$\hat{H}_y(S) = \widehat{\text{HSIC}}(\mathbf{y}, \mathbf{X}|_S), \tag{11}$$

and use a Gaussian kernel. Figure 2 shows the Shapley decomposition of $\hat{H}$ amongst the features generated from Eq. (5), again with $d = 5$ and $A = \text{diag}(0, 2, 4, 6, 8)$. The decomposition has been normalised for comparability with the other measures of dependence presented in the figure. The HSIC can also be generalised to provide a measure of mutual dependence between any finite number of random vectors (*Pfister et al., 2016*).

## EXPLORATION

In machine learning problems, complete formal descriptions of the DGP are often impractical. However, there are advantages to gaining some understanding of the

dependence structure. In particular, such an understanding is useful when *inference* about the data generating process is desired, such as in the contexts of causal inference, scientific inquiries (in general), or in qualitative investigations (cf. *Navarro, 2018*). In a regression or classification setting, the dependence structure between the features and response is an immediate point of focus. As we demonstrate in "Recognising dependence: Example 2", the dependence structure cannot always be effectively probed by computing measures of dependence between labels and feature subsets, even when the number of marginal contributions is relatively small. In such cases, the Shapley value may not only allow us to summarise the interactions from many marginal contributions, but also to fairly distribute strength of dependence to the features.

Attributed dependence on labels (ADL) can be used for exploration in the absence of, or prior to, a choice of model; but, ADL can also be used in conjunction with a model—for example, to support, and even validate, model explanations. Even when a machine learning model is not parsimonious enough to be considered explainable, stakeholders in high risk settings may depend on the statement that "feature $X_i$ is important for determining $Y$" in general. However, it is not always clear, in practice, whether such a statement about feature importance is being used to describe a property of the model, or a property of the DGP. In the following example, we demonstrate that ADL can be used to make statements about the DGP and to help qualify statements about a model.

### Recognising dependence: Example 2

Consider a DGP involving the XOR function of two binary random variables $X_1, X_2$, with distributions given by $P(X_1 = 1) = P(X_2 = 1) = 1/2$. The response is given by

$$Y = \text{XOR}(X_1, X_2) = X_1(1 - X_2) + X_2(1 - X_1). \tag{12}$$

Notice that $P(Y = i | X_k = j) = P(Y = i)$, for all $i, j \in \{0, 1\}$ and $k \in \{1, 2\}$. Thus, in this example, $Y$ is completely statistically independent of each individual feature. However, since $Y$ is determined entirely in terms of $(X_1, X_2)$, it is clear that $Y$ is statistically dependent on the pair. Thus, the features individually appear to have little impact on the response, yet together they have a strong impact when their mutual influence is considered.

Faced with a sample from $(Y, X_1, X_2)$, when the DGP is unknown, a typical exploratory practice is to take a sample correlation matrix to estimate $\text{Cor}(Y, X)$, producing all pairwise sample correlations as estimates of $\text{Cor}(Y, X_i)$, for $i \in [d]$. A similar approach, in the presence of suspected non-linearity, is to produce all pairwise distance correlations, or all pairwise HSIC values, rather than all pairwise correlations. Both the above approaches are model-independent. For comparison, consider a pairwise model-dependent approach: fitting individual single-feature models $M_i$, for $i \in [d]$, that each predict $Y$ as a function of one feature $X_i$; and reporting a measure of model performance for each of the $d$ models, standardised by the result of a *null feature model*—that is, a model with no features (that may, for example, guess labels completely at random, or may use empirical moments of the response distribution to inform its guesses, ignoring $X$ entirely).

As demonstrated by the results in Table 1, it is not possible for pairwise methods to capture interaction effects and mutual dependencies between features. However, Shapley

**Table 1  Importances of features $X_1$ and $X_2$ assigned by various methods, using a sample size of 10,000 from DGP Eq. (12).** For pairwise XGBoost, we take the difference in mean squared prediction error between each XGBoost model and the null model (which always guesses 1). Pairwise dependence includes pairwise DC, HSIC, AIDC and Pearson correlation, which all give the same result of 0, due to statistical independence.

| Method | Result $X_1$ | Result $X_2$ |
| --- | --- | --- |
| SHAP | 3.19 | 3.19 |
| Shapley DC | 0.265 | 0.265 |
| Shapley AIDC | 0.265 | 0.265 |
| Shapley HSIC | 0.16 | 0.16 |
| Pairwise XGB | 0 | 0 |
| Pairwise dependence | 0 | 0 |
| XGB feature importance | 1 | 0 |

feature attributions can overcome this limitation, both in the case of Sunnies and in the case of SHAP. By taking an exhaustive permutations based approach, Shapley values are able to effectively deal with partial dependencies and interaction effects amongst features. Note, all the Sunnies marginal contributions can, in this example, be derived from Table 1: the pairwise results state that $\hat{D}_Y(\{1\}) = \hat{D}_Y(\{2\}) = \hat{D}'_Y(\{1\}) = \hat{D}'_Y(\{2\}) = \hat{H}_Y(\{1\}) = \hat{H}_Y(\{2\}) = 0$, and from Table 1 we can also derive $\hat{D}_Y(\{1,2\}) = \hat{D}'_Y(\{1,2\}) = 2 \times 0.265 = 0.53$ and $\hat{H}_Y(\{1,2\}) = 2 \times 0.16 = 0.32$.

The discrete XOR example demonstrates that ADL captures important symmetry between features, while pairwise methods fail to do so. The results in the final two rows of Table 1 are produced as follows: we train an XGBoost classifier on the discrete XOR problem in Eq. (12). Then, to ascertain the importance of each of the features $X_1$ and $X_2$, in determining the target class, we use the XGBoost "feature importance" method, which defines a feature's *gain* as "the improvement in accuracy brought by a feature to the branches it is on" (see https://xgboost.readthedocs.io/en/latest/R-package/discoverYourData.html).

Common experiences from users suggest that the XGBoost feature importance method can be unstable for less important features and in the presence of strong correlations between features (see e.g., https://stats.stackexchange.com/questions/279730/). However, in the current XOR example, features $X_1$ and $X_2$ are statistically independent (thus uncorrelated) and have the maximum importance that two equally important features can share (that is, together they produce the response deterministically).

Although the XGBoost classifier easily achieves a perfect classification accuracy on a validation set, the associated XGBoost gain for $X_1$ is $\text{Gain}(X_1) \approx 0$, while $\text{Gain}(X_2) \approx 1$, or vice versa. In other words the full weight of the XGBoost feature importance under XOR is given to either one or the other feature. This is intuitively misleading, as both features are equally important in determining XOR, and any single one of the two features is alone not sufficient to achieve a classification accuracy greater than random guessing. In practice, ADL can help identify such flaws with other model explanation methods.

# DIAGNOSTICS

In the following diagnostics sections, we present results using the distance correlation. However, similar results can also be obtained using the HSIC and the AIDC.

## Model attributed dependence

Given a fitted model $f$, with associated predictions $\hat{Y} = f(X)$, we seek to attribute shortcomings of the fitted model to individual features. We can do this by calculating the Shapley decomposition of the estimated strength of *dependence* between the model residuals $\varepsilon = Y - \hat{Y}$, and the features $X$. In other words, feature $v$ receives the attribution $\phi_v(C_\varepsilon)$; estimated by $\phi_v(\hat{C}_\mathbf{e})$, where $\mathbf{e} = \mathbf{y} - \hat{\mathbf{y}}$. We refer to this as the Attributed Dependence on Residuals (ADR) for feature $v$.

A different technique, for diagnosing model misspecification, is to calculate the Shapley decomposition of the estimated strength of dependence between $\hat{Y}$ and $X$, so that each feature $v$ receives attribution $\phi_v(\hat{C}_{\hat{y}})$. We call this the Attributed Dependence on Predictions (ADP), for feature $v$. This picture of the model generated dependence structure may then be compared, for example, to the observed dependence structure given the ADL $\{\phi_v(\hat{C}_\mathbf{y}) : v \in [d]\}$. The diagnostic goal, then, may be to check that, for all $v$,

$$|\phi_v(\hat{C}_{\hat{y}}) - \phi_v(\hat{C}_\mathbf{y})| < \delta, \tag{13}$$

for some $\delta$ tolerance. In other words, a diagnostic strategy making use of ADP is to compare estimates of feature importance under the model's representation of the joint distribution, to estimates of feature importance under the empirical joint distribution, and thus to individually inspect each feature for an apparent change in predictive relevance.

We note that these techniques, ADP and ADR, are agnostic to the chosen model. All that is needed is the model outputs and the corresponding model inputs—the inner workings of the model are irrelevant for attributing dependence on predictions and residuals to individual features in this way.

### Demonstration with concept drift

We illustrate the ADR and ADL techniques together with a simple and intuitive synthetic demonstration involving concept drift, where the DGP changes over time, impacting the mean squared prediction error (MSE) of a deployed XGBoost model. The model is originally trained with the assumption that the DGP is static, and the performance of the model is monitored over time with the intention of detecting violations of this assumption, as well as attributing any such violation to one or more features. A subset of the deployed features can be selected for scrutiny, by considering removal only of those selected features from the model. To highlight this, our simulated DGP has 50 features, and we perform diagnostics on 4 out of those 50 features.

For comparison, we compute SAGE values of the model mean squared error (*Covert, Lundberg & Lee, 2020*) and we compute the mean SHAP values of the logarithm of the model loss function (*Lundberg et al., 2020*). We will refer to the latter as SHAPloss. For SAGE and SHAPloss values, we employ a DGP similar to Eq. (14), with sample size 1000, but with $X_i \equiv 0$ for all $i > 4$. These features were nullified for tractability of the SAGE

computation, since, unlike for Sunnies, the authors are not aware of any established method for selectively computing SAGE values of a subset of the full feature set. SAGE and SHAPloss were chosen for their popularity and ability to provide *global* feature importance scores.

At the initial time $t = 0$, we define the DGP as a function of temporal increments $t \in \mathbb{N} \cup \{0\}$,

$$Y = X_1 + X_2 + \left(1 + \frac{t}{10}\right) X_3 + \left(1 - \frac{t}{10}\right) X_4 + \sum_{i=5}^{50} X_i, \tag{14}$$

where $X_i \sim N(0, 4)$, for $i = 1, 2, 3, 4$, and $X_i \sim N(0, 0.05)$, for $5 \leq i \leq 50$. Features 1 through 4 are the most effectual to begin with, and we can imagine that these were flagged as important during model development, justifying the additional diagnostic attention they enjoy after deployment. We see from Eq. (14) that, after deployment, i.e., during periods $1 \leq t \leq 10$, the effect of $X_4$ decreases linearly to 0, while the effect of $X_3$ increases proportionately over time. In what follows, these changes are clearly captured by the residual and response dependence attributions of those features, using the DC characteristic function.

The results, with a sample size of $n = 1000$, from the DGP in Eq. (14), are presented in Fig. 3. According to the ADL (top), $X_4$ shows early signs of significantly reduced importance $\phi_4(\hat{C}_\mathbf{y})$, as $X_3$ shows an increase in importance $\phi_3(\hat{C}_\mathbf{y})$, which is roughly symmetrical to the decrease in $\phi_4(\hat{C}_\mathbf{y})$. The ADR (bottom) show early significant signs that $X_3$ is disproportionately affecting the residuals, with high $\phi_3(\hat{C}_\mathbf{e})$. The increase in residual attribution $\phi_4(\hat{C}_\mathbf{e})$ is also evident, though the observation $\phi_4(\hat{C}_\mathbf{e}) < \phi_3(\hat{C}_\mathbf{e})$ suggests that the drift impact from $X_3$ is the larger of the two.

The resulting SAGE and mean SHAPloss values are presented in Fig. 4. Interestingly, the behaviours of SHAPloss and SAGE are (up to scale and translation) analogous to the behaviour of ADL, rather than ADR, despite the model-independence of ADL. A reason for this, in this example, is that the feature with higher (resp. lower) dependence on $Y$ contributes less (resp. more) to the residuals. When interpreting the SHAPloss and SAGE outputs, it is important to note that the model loss is increasing with $t$, since the true planar trend in $(Y, X_3, X_4)$ rotates away from the learned trend. So, while the SAGE and SHAPloss results may appear to make the paradoxical suggestion that $X_3$ is utilised better by the model at $t = 10$ compared with $t = 0$, this is not the case: SAGE and SHAPloss are not accounting for the change in model loss over time. The model loss may decrease more when marginalising a feature under a misspecified model, than under a model with lower overall loss.

### Demonstration with misspecified model

To illustrate the ADL, ADP and ADR techniques, we demonstrate a case where the model is misspecified on the training set, due to model bias. The inadequacy of this misspecified model is then detected on the validation set. Unlike the example given in "Model attributed dependence", the DGP is unchanging between the two data sets. The key technique used in this demonstration is the comparison of differences between ADL (calculated in the absence of any model) and ADP (calculated using the output of a fitted model), in order to identify

**(A)**

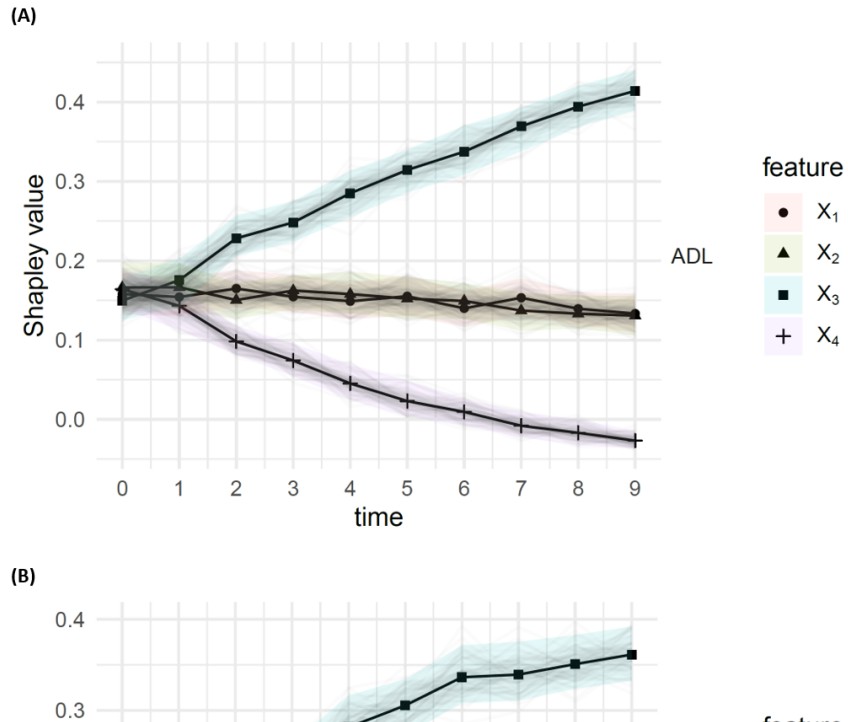

**(B)**

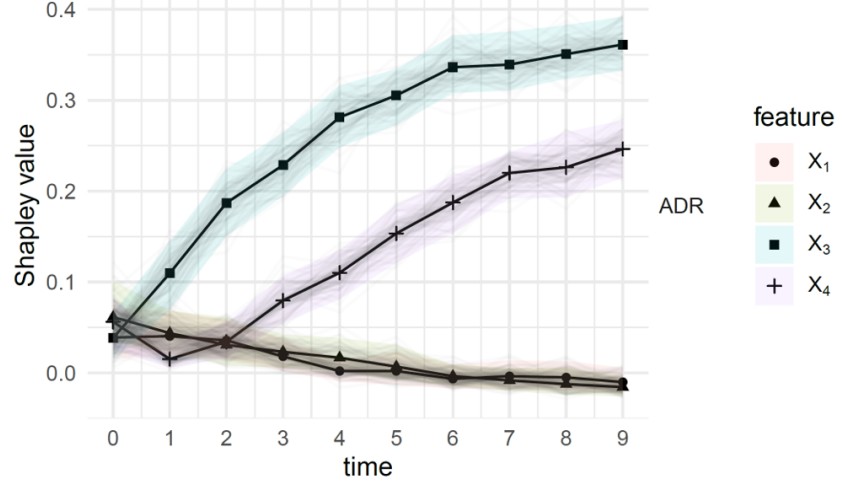

**Figure 3** **Attributed dependence on (A) labels (ADL) and (B) residuals (ADR), using the DC characteristic function; results for times $t \in \{0, \dots, 10\}$, from a simulation with sample size 1,000 from the DGP Eq. (14).** The bootstrap confidence bands are the 95% middle quantiles ($Q_{0.975} - Q_{0.025}$) from 100 subsamples of size 1,000. The ADL of features $X_3$ and $X_4$ appear to decay/increase over time, leading to significantly different ADL, compared to the other features. We also see that $X_3$ and $X_4$ have significantly higher ADR than the other features.

any differences in the attributions between dependence on labels and the dependence on the predictions produced by the misspecified model. Such a comparison, between model absence and model outputs, is not possible using purely model-dependent Shapley values.

To make this example intuitive, we avoid using a complex model such as XGBoost, in favour of a linear regression model. Since the simulated DGP is also linear, this example allows a simple comparison between the correct model and the misspecified model. The

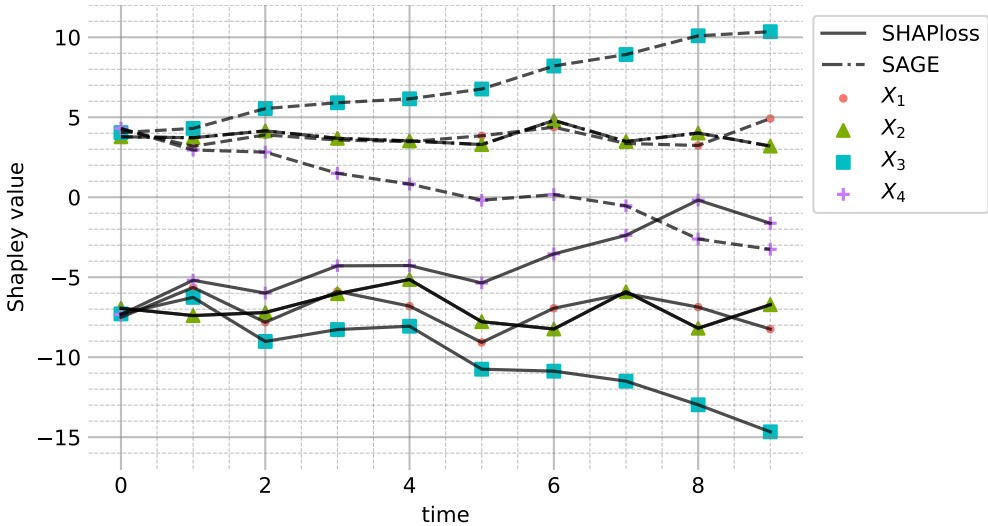

**Figure 4  SAGE and SHAPloss values for the modified simulated DGP Eq. (14) (see "Demonstration with concept drift"), with sample size 1,000.** The results are comparable to ADL (see Fig. 3). Features with high SAGE value contribute less to residuals, and vice versa for the SHAPloss values.

DGP in this example is

$$Y = X_1 + X_2 + 5X_3X_4X_5 + \varepsilon, \tag{15}$$

where $X_1, X_2, X_3 \sim N(0,1)$ are continuous, $\varepsilon \sim N(0,0.1)$ is a small random error, and $X_4, X_5 \sim \text{Bernoulli}(1/2)$ are binary. Hence, we can make the interpretation that the effect of $X_3$ is modulated by $X_4$ and $X_5$, such that $X_3$ is effective, only if $X_4 = X_5 = 1$. For this demonstration, we fit a misspecified linear model $EY = \beta_0 + \beta X$, where $X^T = (X_i)_{i=1}^d$ is the vector of features, and $\beta_0, \beta = (\beta_i)_{i=1}^d$ are real coefficients. This is a simple case where the true DGP is unknown to the analyst, who therefore seeks to summarise the 80 marginal contributions from 5 features into 5 Shapley values.

Figure 5 shows the outputs for attributed dependence on labels, residuals and predictions, via ordinary least squares estimation. From these results, we make the following observations:

(i)   For $X_3$ the ADP is significantly higher than the ADL.

(ii)  For $X_4$ and $X_5$ the ADP is significantly lower than the ADL.

(iii) For $X_1, X_2$ there is no significant difference between ADP and ADL.

(iv)  For $X_1, X_2$ ADR is negative, while $X_3, X_4, X_5$ have positive ADR.

Observations (i) and (ii) suggest that the model $EY = \beta_0 + \beta X$ overestimates the importance of $X_3$ and underestimates the importance of $X_4$ and $X_5$. Observations (iii) and (iv) suggest that the model may adequately represent $X_1, X_2, X_3$, but that $X_3, X_4$ and $X_5$ are significantly more important for determining structure in the residuals than $X_1$ and $X_2$. A residuals versus fits plot may be useful for confirming that this structure is present and of large enough magnitude to be considered relevant.

Having observed the result in Fig. 5, for the misspecified linear model $EY = \beta_0 + \beta X$, we now fit the correct model: $EY = \beta_0 + \beta X + X_3X_4X_5$, which includes the three-way

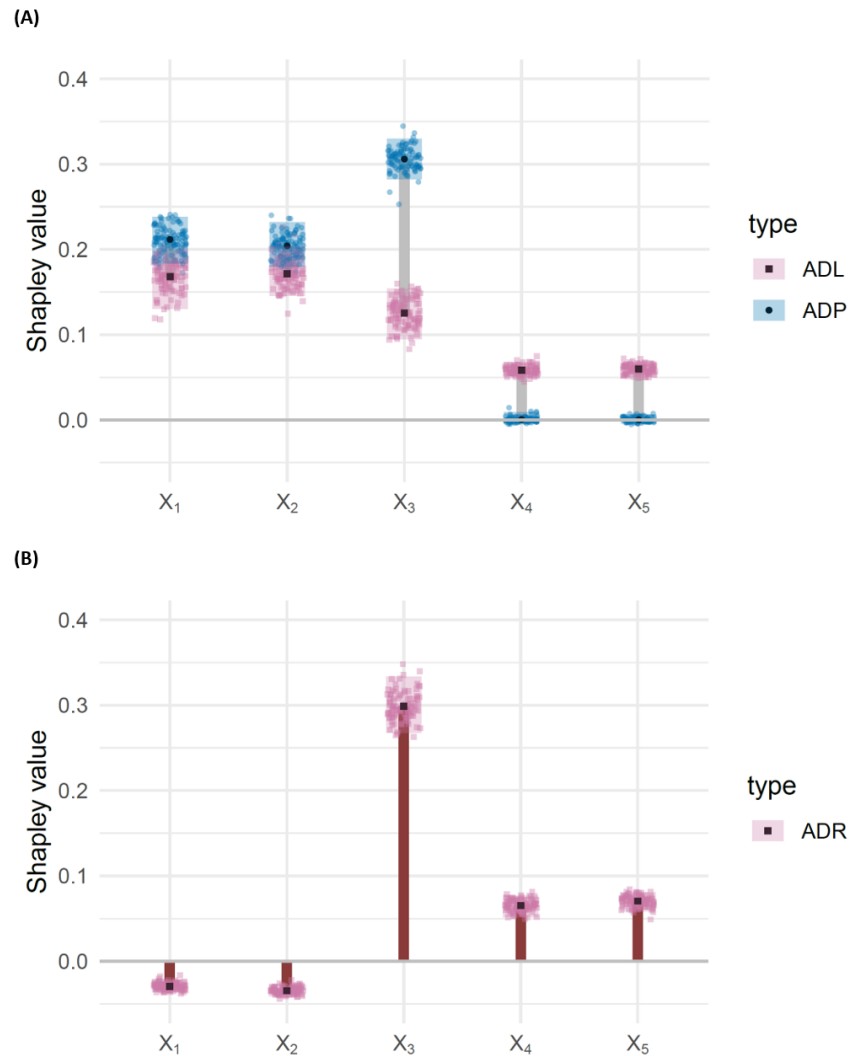

**Figure 5** (A) Barbell plots showing differences in attributed dependence on labels (ADL), based on the DC characteristic function between the training and test sets, for each feature, for the misspecified model $EY = \beta_0 + \beta X$ with DGP (Eq. (15)). Larger differences indicate that the model fails to capture the dependence structure, effectively. (B) Bar chart representing attributed dependence on residuals (ADR) for the test set. The shaded rectangles represent bootstrap confidence intervals, taken as the 95% middle quantile ($Q_{0.975} - Q_{0.025}$) from 100 resamples of size 1,000. Non-overlapping rectangles indicate significant differences. Point makers represent individual observations from each of the 100 resamples.

interaction effect $X_3X_4X_5$. The results, shown in Fig. 6, show no significant difference between the ADL and ADP for any of the features, and no significant difference in ADR between the features.

## APPLICATION TO DETECTING GENDER BIAS

We analyse a mortality data set produced by the US Centers for Disease Control (CDC) via the National Health and Nutrition Examination Survey (NHANES I) and the NHANES I Epidemiologic Follow-up Study(NHEFS) (*Cox, 1998*). The data set consists of 79 features

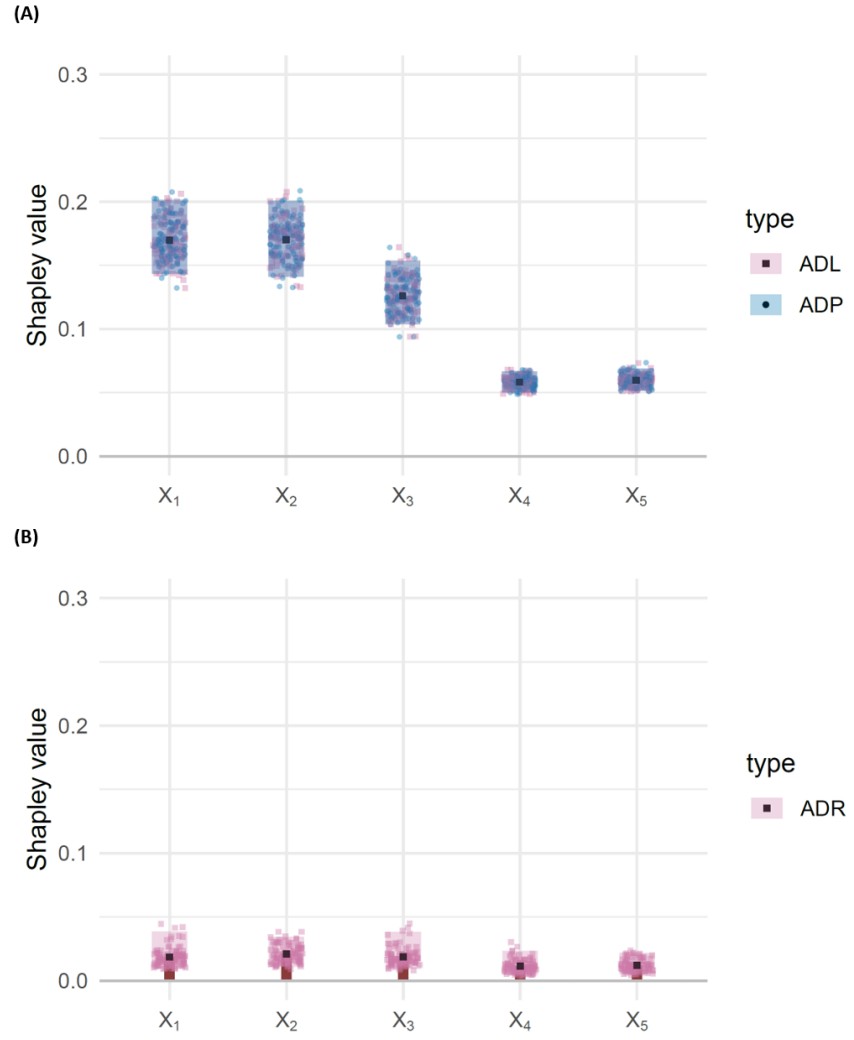

**Figure 6** (A) Barbell plots showing differences in attributed dependence on labels (ADL), based on the DC characteristic function between the training and test sets, for each feature, for the correctly specified model with DGP (Eq. (15)). The shaded rectangles represent bootstrap confidence intervals, taken as the 95% middle quantile ($Q_{0.975} - Q_{0.025}$) from 100 resamples of size 1,000. Overlapping rectangles indicate non-significant differences, suggesting no evidence of misspecification. Point markers represent individual observations from each of the 100 resamples. (B) Bar chart representing attributed dependence on residuals (ADR) for the test set. Compare to Fig. 5.

from medical examinations of 14,407 individuals, aged between 25 and 75 years, followed between 1971 and 1992. Amongst these people, 4,785 deaths were recorded before 1992. A version of this data set was also recently made available in the SHAP package (*Lundberg & Lee, 2017*). The same data were recently analysed in Section 2.7, *Lundberg et al. (2020)* (see also https://github.com/suinleelab/treeexplainer-study/tree/master/notebooks/mortality).

We use a Cox proportional hazards objective function in XGBoost, with learning rate (eta) 0.002, maximum tree depth 3, subsampling ratio 0.5, and 5,000 trees. Our training set contain 3,370 observations, balanced via random sampling to contain an equal number of

**Table 2  The 16 features used for fitting a Cox proportional hazards model to NHANES I and NHEFS data.**

| Feature name | Feature name |
| --- | --- |
| Age | Sex |
| Race | Serum albumin |
| Serum cholesterol | Serum iron |
| Serum magnesium | Serum protein |
| Poverty index | Physical activity |
| Red blood cells | Diastolic blood pressure |
| Systolic blood pressure | Total iron binding capacity |
| Transferrin saturation | Body mass index |

males and females. We then test the model on three different data sets: a all male test set of size 1686, containing all males not in the training data; an all female test set of size 3,547, containing all females not in the training data; and a gender balanced test set of size 3,372. The data are labelled with the observed time-to-death of each patient during the follow-up study. For model fitting, we use the 16 features given in Table 2.

Of the features in Table 2, we focus on the Shapley values for a subset of well-established risk factors for mortality: age, physical activity, systolic blood pressure, cholesterol and BMI. Note that the results presented here are purely intended as a proof of concept—the results have not been investigated in a controlled study and none of the authors are experts in medicine. We do not intend for our results to be treated as a work of medical literature.

We decompose dependence on the labels, model predictions and residuals, amongst the three features: age, systolic blood pressure (SBP) and physical activity (PA), displaying the resulting ADL, ADP and ADR for each of the three test data sets in Fig. 7(using the DC characteristic function). From this analysis we make the following observations.

(i)  Age has a significantly higher attributed dependence on residuals compared with each of the other features, across all three test sets. This suggests that age may play an important role in the structure of the model's residuals. This observation is supported by the dumbbells for age, which suggest a significant and sizeable difference between attributed dependence on prediction and attributed dependence on labels; that is, we have evidence that the model's predictions show a greater attributed dependence on age than the labels do.

(ii)  For SBP, we observe no significant difference between ADL and ADP for the balanced and all male test sets. However, in the all female test set, we do see a significant and moderately sized reduction in the attributed dependence on SBP for the model's predictions compared with that of labels. This suggests that the model may represent the relationship between SBP and log relative risk of mortality less effectively on the all female test set than on the other two test sets. This observation is supported by the attributed dependence on residuals for SBP, which is significantly higher in the all female test set compared to the other two sets.

(iii)  For PA, we see a low attributed dependence on residuals, and a non-significant difference between ADL and ADP, for all three test sets. Thus we do not have any

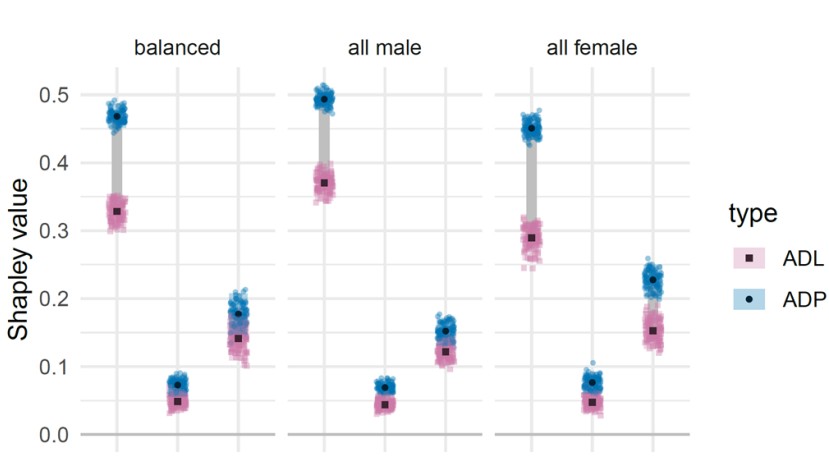

**Figure 7** **(A) Shapley decomposition of attribution dependence on labels (ADL, pink) and predictions (ADP, blue), and (B) residuals (ADR, orange) for the three features age, physical activity (PA) and systolic blood pressure (SBP), on three different test data sets consisting of an equal proportion of females and males ("balanced"), only male ("all male") and only females ("all female").** . Attributes were based on the DC characteristic function.

reason, from this investigation, to suspect that the effect of physical activity is being poorly represented by the model.

The results regarding potential heterogeneity due to gender and systolic blood pressure are not suprising given that we expect, *a priori*, there to be a relationship between systolic blood pressure and risk of mortality (*Port et al., 2000*), and that studies also indicate this relationship to be non-linear (*Boutitie et al., 2002*), as well as dependent on age and gender (*Port, Garfinkel & Boyle, 2000*). Furthermore, the mortality risk also depends on age and gender, independently of blood pressure (*Port, Garfinkel & Boyle, 2000*). We also expect physical activity to be important in predicting mortality risk (*Mok et al., 2019*).

# DISCUSSION AND FUTURE WORK

After distinguishing between model-dependent and model-independent Shapley values, in "Measures of non-linear dependence", we introduce energy distance-based and kernel-based characteristic functions, for the Shapley game formulation, as measures of non-linear dependence. We assign the name 'Sunnies' to Shapley values that arise from such measures.

In "Recognising dependence: Example 1" and "Exploration", we demonstrate that the resulting model-independent Shapley values provide reasonable results compared to a number of alternatives on certain DGPs. The alternatives investigated are the XGBoost built-in feature importance score, pairwise measures of non-linear dependence, and the $R^2$ characteristic function. The investigated DGPs are a quadratic form, for its simple non-linearity; and an XOR functional dependence, for its absence of pairwise statistical dependence. These examples are simple but effective, as they act as counter-examples to the validity of the targeted measures of dependence to which we draw comparison.

In "Diagnostics", we demonstrate how the Shapley value decomposition, with these non-linear dependence measures as characteristic function, can be used for model diagnostics. In particular, we see a variety of interesting examples, where model misspecification and concept drift can be identified and attributed to specific features. We approach model diagnostics from two angles, by scrutinising two values: the dependence attributed on predictions by the model (ADP), and the dependence between the model residuals and the input features (ADR). These are proofs of concept, and the techniques of attributed dependence on labels (ADL), ADP and ADR require development to become standard tools. However, the examples highlight the techniques' potential, and we hope that this encourages greater interest in them.

We provide two demonstrations of the diagnostic methods: in "Demonstration with concept drift", we use a data generating process which changes over time, and where the deployed model was trained at one initial point in time. Here, Sunnies successfully uncovers changes in the dependence structures of interest, and attributes them to the correct features, early in the dynamic process. The second demonstration, in "Demonstration with misspecified model", shows how we use the attributed dependence on labels, model predictions and residuals, to detect which features' dependencies or interactions are not being correctly captured by the model. Implicit in these demonstrations is the notion that the information from many marginal contributions is being summarised into a human digestible number of quantities. For example, in "Demonstration with misspecified model", the 80 marginal contributions from 5 features are summarised as 5 Shapley values, in each of ADL, ADP and ADR, facilitating the simple graphical comparison in Fig. 5.

There is a practical difference between model-independent and model-dependent methods, highlighted in "Demonstration with misspecified model", when comparing the dependence structure in a data set, to the dependence structure captured by a model. Model-independent methods can be applied to model predictions and residuals, but can also be applied to data labels as well. Thus, techniques using model-independent Shapley values will be markedly different from model-dependent methods in both design and interpretation. Indeed, consider that there is a different interpretation between (a)

the decomposition of a measure of *statistical* dependence, e.g., as a measure of distance between the joint distribution functions, with and without the independence assumption, and (b) the attribution of a measure of the *functional* dependence of a model on the value of its inputs.

While the DC does provide a population level (asymptotic) guarantee that dependence will be detected, it must be noted that, as discussed in "Distance correlation and affine invariant distance correlation", the DC tends to be greater for a linear association than for a non-linear association. These are not strengths, or weaknesses, of using a measure of non-linear statistical dependence as the Shapley value characteristic function (i.e., the method we call Sunnies) but rather of the particular choice of characteristic function in this method. Work is needed to investigate other measures of statistical dependence in place of DC, HSIC or AIDC, and to provide a comparison between these methods, including a detailed analysis of strengths, limitations and computational efficiency. In this paper, we have not focused on such a detailed experimental evaluation and comparison, but on the exposition of the Sunnies method itself. A potential alternative to our use of energy correlation and HSIC is the class of maximal information based non-parametric exploration (MINE) statistics, or other mutual information based measures (*Kinney & Atwal, 2014*; *Reshef et al., 2011*).

Finally, in "Application to Detecting Gender Bias", we apply Sunnies to a study on mortality data, with the aim of detecting effects caused by gender differences. We find that, when the model is trained on a gender balanced data set, a significant difference is detected between the model's representation of the dependence structure via its predictions (ADP) and the dependence structure on the labels (ADL); a difference which is significant for females and not for males, even though the training data was gender balanced. Although we do not claim that our result is causal, it does provide evidence regarding the potential of Sunnies to uncover and attribute discrepancies that may otherwise go unnoticed, in real data.

A well-known limitation when working with Shapley values, is their exponential computational time complexity. Ideally, in "Application to Detecting Gender Bias", we would have calculated Shapley values of all 17 features. However, it is important to note that we *do not need* to calculate Shapley values of all features, if there is prior knowledge available regarding interesting or important features, or if features can be partitioned into independent blocks. To illustrate the idea of taking advantage of independent blocks, suppose we have a model with 15 features. If we know in advance that these features partition into 3 independent blocks of 5 features, then we can decompose the pairwise dependence of each block into 5 Shapley values. In this way, 15 Shapley values are computed from 240 within-block marginal contributions, rather than the full number of $32,768$ marginal contributions. In the future, it may be interesting to also consider the computational efficiencies that may arise in scenarios where the sparsity structures of marginal contributions can be directly exploited, as well as the potential for examining such marginal contributions directly (e.g., via visualisation).

Finally, note that we have made the distinction that Shapley feature importance methods may or may not be model-dependent, but this distinction holds for model explanation

methods in general. We believe that complete and satisfactory model explanations should ideally include a description from both categories.

All code and data necessary to produce the results in this manuscript are available on github.com/ex2o/sunnies.

### Funding
The authors received no funding for this work.

### Competing Interests
The authors declare there are no competing interests.

### Author Contributions
- Daniel Vidali Fryer and Inga Strumke conceived and designed the experiments, performed the experiments, analyzed the data, performed the computation work, prepared figures and/or tables, authored or reviewed drafts of the paper, and approved the final draft.
- Hien Nguyen conceived and designed the experiments, authored or reviewed drafts of the paper, and approved the final draft.

### Data Availability
Data, code and scripts for generating simulated data are available at GitHub: https://github.com/ex2o/sunnies.

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
