# Peer review of "Model independent feature attributions: Shapley values that uncover non-linear dependencies"

_PeerJ Computer Science, doi:10.7717/peerj-cs.582_

## Round 0.1 · original submission · Major Revisions

Please address the suggestions and questions from reviewers thoroughly. Please also write a separate letter listing and explaining your changes, in addition to improving the paper itself.

·

Basic reporting

+ Couple of typos it seems.
Line 250: Delete X3.
Line 296: Change Age to SBP.

+ Paragraph starting at line 63 can be improved.
We know computational complexity is a big issue using Shapley values. Discussed at lines 63 and 343.
If proposed Sunnies can use any of the ideas proposed for efficient computation, state it. If not, also state it explicitly, saying future work is needed for larger data sets.

Experimental design

+ Equation (5) with d=4 and A=diag(0,2,4,6) is used for figure 2. But there are five features in figure 2. I think there is a typo somewhere.

+ Concept drift experiment (line 228) can use some more context.
Comparison methods might help, e.g. XGBoost feature importance (which you can say is model-dependent), or pairwise correlations (linear or non-linear).
Or maybe state that clearly a linear (or non-linear) pairwise correlation would do here, but the point here is that your method is general. I might be overthinking this!

Validity of the findings

Please bear with me while I elaborate more on complexity issue again.
Line 343 you mention it is useful to partition features to independent blocks which is technically true.
But a big question arises, if researchers in a domain can partition features to (non-interacting) independent blocks (of size of about five features), why not just study non-linear independence relations (using DC, AIDC, or HSIC), or even more traditional, why not try interaction terms in regression models and check likelihood scores? After all, there are only handful of features in each independent block.
Let me put it another way. If we only have small number of features
why not look at each term C(SU{v})-C(S) separately, (which probably you've saved when calculating non-linear Shapley values), rather than adding these up to get Shapley value and lose the local information? i.e. simply using non-linear correlations you listed, but not doing the summation over subsets S and reporting single value per feature.

Additional comments

+ Example 2 (line 175). I think it'd be informative to also list DC, AIDC, HSIC among {X1}, {X2},
{X1, X2} and Y.

+ To convince the audience that non-linear Shapley values are helpful, you kind of have to show them at least one example that the number of features is high and simply looking at a list of DC, AIDC, or HSIC is not practical.

+ The misspecified model (line 241) is a very clear and essential example! Here one has to move away from (pairwise or linear) and consider ("more global" and non-linear) hence your non-linear Shapley.
But again, this example is also too small! One can look at the list of nonlinear correlations rather than adding them up into Shapley.

+ (Plug alert!) In our paper (S. Ma, -- 2020), we showed that Shapley values can be misleading for learning local causal neighborhoods, using two simple examples of Markovian graphs (i.e. faithfullness assumption were met).
The general statement (which is our current belief) is related to the issues above. Basically, adding up all C(SU{v})-C(S) terms, one loses the important local information about v and certain S. At least for causal discovery, Shapley's way of adding C(SU{v})-C(S) terms, though mathematically attractive with additvity on C and all, seems arbitrary, or I shall say, counterproductive.

·

Basic reporting

No comment.

Experimental design

I like the proposed method but believe it needs a much more rigorous investigation ("Rigorous investigation performed to a high technical & ethical standard). There are several experiments where Shapley value based, model-dependent methods exist for the chosen problems and should be compared against. It is not necessary, especially given PeerJ's explicit willingness to publish negative or inconclusive results, to outperform these other methods, but it is necessary to include them in experiments.

3.0.1 - Here a decision tree perfectly learns the XOR function but the built-in feature importances do not correctly assign importance. The take-away message seems to be that model-dependent explanations don't correctly handle interactions like XOR or at least that they are flawed in scenarios like this one, and I don't think this is true. The authors cite TreeExplainer and SHAP, and the SHAP repository actually uses XOR as an example where XGBoost's built-in importance does not correctly assign credit but TreeSHAP/TreeExplainer does. So I think it's necessary to include a comparison to TreeSHAP/TreeExplainer here (which should correctly assign credit).
https://shap.readthedocs.io/en/latest/example_notebooks/tree_explainer/Understanding%20Tree%20SHAP%20for%20Simple%20Models.html
Minor comment - I don't think it's ever explained what the "X1" and "X2" columns of Table 1 are -- presumably the importances of X1 and X2 under various methods.

4.1.1 - Model monitoring and explaining the loss are great experiments; however, this is also done in TreeExplainer, which should be referenced and compared against. This comparison is especially important because TreeExplainer is cited earlier in the paper and tree models are used for this experiment. TreeExplainer demonstrates how to (1) calculate Shapley values of the model's loss and (2) use loss explanations to perform model monitoring in the presence of feature drift. It's hard to justify publishing a section on explaining model mistakes with feature drift in trees without referencing existing experiments that do exactly that. I think the results will be informative and useful regardless of their exact outcome. It is also possible to calculate Shapley values with standard model-agnostic SHAP methods by treating the function to be explained as the model loss rather than the model output, and I think it would be useful (1) try doing so and (2) discuss how the ADR method differs from that approach.

4.1.2 - Some minor comments: Point (iii) should not include X3. Also, I'm not sure lines 261-263 contribute to the main points of the section. Which of the linear models is both accurate and parsimonious? I think this is meant to be the one that includes the correct interaction term. However, the paper doesn't include R^2 or other metrics to compare accuracy of these models, and even if such metrics were present I'm not sure there's an important broader point to be made, since the section isn't really about how to train sparse/accurate ML models.

Overall, the experimental results are interesting, but feel incomplete without including comparisons to some existing methods for answering similar questions.

Validity of the findings

No comment.

Additional comments

I like this paper -- I think the method is a great idea and the paper contributes to important discussions in ML about explaining the model vs explaining the dataset. I do believe there are large experimental gaps in the current version of the paper, and that the paper should not be published without addressing these gaps and acknowledging other important methods (e.g., Shapley values of the model loss) for answering the questions the paper poses. However, if these gaps are adequately addressed and sufficient additional experiments are run, the paper would be a valuable resource for the ML community. My most important concerns are the experimental concerns noted under (2) Experimental Design. Several other questions and concerns:

1 - I am not sure there is enough discussion of why we should consider HSIC, AIDC, etc to be non-parametric and not consider explanations based on a complex model (gradient boosting, deep models, etc) to be non-parametric.
a) Sufficiently complex models are often themselves described as non-parametric.
b) No free lunch theorems, etc, in ML make me think that there is probably not a perfect solution to feature dependence just as there is no perfect solution to prediction. I would probably be more comfortable with an explicit discussion of the pros and cons of model-dependent versus model-independent Shapley methods than with an overly broad claim that model-independent methods make no assumptions about the DGP.
c) Some specific pros and cons that come to mind: a strength of the model-independent methods is the guarantee in 2.2 that dependencies will always be detected. A strength of model-dependent methods is that they may be better suited to measure the *strength* of interaction -- for example, from 2.2.1, distance correlation should counterintuitively be higher for a small linear association than for a large nonlinear one, which would likely not hold for an XGBoost model explanation. More discussion of the value of the chosen properties would be very useful -- maybe there are certain cases where it's particularly important that we never accidentally miss an important feature, in which the guarantee from 2.2 is particularly important. In other cases maybe other properties are more important and other methods should be used.

Overall I am not concerned about whether model-independent explanations are useful -- they definitely seem to be! But I think more discussion of the the assumptions made in this approach and how they differ from the assumptions in model-dependent explanations would be helpful.

2 - I think it would help to explain why we need to use Shapley values with HSIC or AIDC as characteristic functions, rather than simply applying them out of the box. For example, a raw pairwise correlation vector is often used to understand relationships between features and labels "in the data". Why not simply calculate, e.g., AIDC between all features and labels? Why is the Shapley value helpful? Personally, I have an intuition for why marginal effects differ from Shapley values for ML models but don't have a similar intuition in the case of HSIC and AIDC.

3 - In Section 4, ADL is frequently discussed and plotted but it is not clear to me how it is calculated (i.e., with HSIC or AIDC, etc). I may have missed these details but I think it would be helpful to clarify exactly what method is being used in the plots. I think it is essential to clarify this before publication.

---

## Round 0.2 · Minor Revisions

Dear authors:

Thank you for an excellent paper. Reviewer 2 asks for another round of reviewing, but to benefit everyone involved, I do not think that is needed. Nevertheless, please do address the issue "It’s very counterintuitive to me that a feature with substantial drift could have a negative contribution to loss, especially a more negative contribution than features with no drift."

Section 2.2 is about measure of non-independence. The 2011 paper in Science "Detecting Novel Associations in Large Data Sets" attracted a lot of attention then, and now has over 1800 citations. Please consider discussing the relevance of this paper, and/or of papers that improve on it.

Please fix capitalization issues in the references, such as "hilbert-schmidt".

·

Basic reporting

+ Typos,
Equation 1: S_k under summation, the font should be calligraphic.
Line 173: d=5, A=(0,2,4,6,8)

Experimental design

+ Lines 301 and 391. If you have 5 features, wouldn't you have 5*2^4=80 marginal contributions? Maybe put the formula in terms of d.
I think line 426 is correctly 240, because there are 3 independent blocks with size 5 each.

Validity of the findings

No comment.

Additional comments

+ I understand the argument why it is much easier to look at per feature Shapley values. But let me say a few words.
Usually the list of the marginal contributions is sparse or consists of few distinct values. For example, if C({1,2, 3}) - C({1,2})=0 then any S where {1,2} \in S would give zero marginal contribution for 3. Similarly if features are not interacting then you get distinct values. Maybe you can state that after quick diagnostics using only 5 Shapley values one can always go back and look into the marginal terms for further diagnostics. But I understand expanding on this point would probably need another paper.

+ I think lines 139-142 (Section 2.2) can use some clarification. One can use non-linear correlations to figure out visible dependence before summing it up into Shapley values.
For XOR, it's true that even non-linear correlations give zero for X1 and X2, but S={X1,X2} has non-zero non-linear correlation. This is why Shapley is non-zero to begin with. Following from previous point above, having same Shapley value would not tell you if relation is X1+X2, X1*X2, X1 XOR X2, .... But going back and checking where Shapley value came from gives you this information.
...
Just wanna mention that seeing Cy(S) = 0 but Cy(S') nonzero, when S \in S' can happen even in a linear additive causal graphs when observing collider child makes the corresponding spouse[s] dependent.

·

Basic reporting

no comment

Experimental design

Overall I think the changes have addressed almost all of my concerns and there is only one major point I’d like to see addressed before I’d recommend publication. I checked "major revisions" because I would ideally like to see the revisions in response to this point, but I do not think it will be a huge amount of work to address.

Major point:
In section 4.1.1, I need to understand better why SHAPloss more closely matches the ADL plot rather than ADR. It’s very counterintuitive to me that a feature with substantial drift could have a negative contribution to loss, especially a more negative contribution than features with no drift. I feel uncomfortable accepting the paper without a bit more understanding of going on here.

The necessary revision here could consist of extra experiments or simply a thoughtful discussion of why the phenomenon might be occurring, to show the authors have covered their bases and this isn’t an implementation problem. One way I thought this could be addressed is by training a linear model rather than a tree model; since the DGP is linear, there’s always the possibility something weird is happening with the learned tree model. Running SAGE with a trained linear model could be a helpful point of comparison. The counterintuitive findings may persist, and that's OK -- even good -- but overall, I'd just like to see a little more investigation and/or discussion of this point.

Minor point:
It’s a little unintuitive in Figure 2 that feature 1 attribution isn’t 0. This is likely because of normalization, but maybe we could have a second figure to show unnormalized Shapley values? Or some other explanation of why feature 1 doesn’t have 0 attribution.

Validity of the findings

no comment

Additional comments

Minor notes:
Is it possible to make references to Figure 2 clearer? Section 2.2.1 doesn’t reference it at all, and 2.2.2 only a little. It would be good to see a comprehensive discussion of the figure, maybe in a new subsection 2.2.3? This doesn’t need to have much more material; it would just clarify things if 2.2.1 and 2.2.2 were exclusively for defining metrics, then 2.2.3 discussed (a) how Shapley values were normalized, (b) how HSIC/DC/AIDC results differ from R2 results, (c) any discussion you have on the differences among HSIC/DC/AIDC results. The only absolutely necessary new thing to include would be the discussion of how normalization occurred.

In Section 3 a little more analysis of the SHAP and Sunnies results could be useful. They both seem to work well, but as you mentioned earlier in the paper, it’s interesting that Sunnies gets results exactly as good as SHAP without ever training a model in the first place! There are many potential benefits, but one that I think should be discussed here is that many model classes have a lot of trouble learning XORs. Thus, not depending on the model learning this kind of interaction is an advantage. (Optionally, you could potentially even show that SHAP with a scikit-learn decision tree model doesn’t correctly learn importance for X1 and X2 because it can't even learn XOR - thus, SHAP's success here depends on learning a good model whereas Sunnies don't).

In section 5, another potential benefit of Sunnies is that ADR type metrics could be very hard to calculate for Cox loss; right now I believe TreeExplainer supports explaining MSE and binary log-losses, but doesn’t support Cox. This could be very hard to implement because unlike other losses, Cox loss for a single sample can depend on all other labels in the dataset! Having a method for residual explanations that doesn’t depend on the loss having a specific convenient form could be very useful. (It is of course possible to calculate Cox loss explanations using model-agnostic SHAP methods or possibly SAGE; whether these are preferable to Sunnies could come down to runtime, fidelity to the model, or other considerations).

Maybe also worth mentioning that the barbell plot method for Sunnies supports a valuable modeling loop that addresses issues in both supervised learning and model-free HSIC type methods. Generally, supervised models run the risk of being misspecified, which is a drawback. Methods like HSIC, DC, AIDC claim to always be able to detect dependence, but they can’t tell you the form of that dependence, which is a drawback. Sunnies help fill both gaps by iteratively training and tweaking your model until ADL and ADP match well. (Main point here that’s not in the paper is that the lack of an ability to make predictions is a drawback for the otherwise nice property of methods like HSIC).

Lines 371-72, should mention SHAP is also an alternative -- Sunnies get reasonable results compared to SHAP as well (i.e., the same results).

Line 381 mentions ADL but doesn’t define it like ADR and ADP.

---

## Round 0.3 · accepted · Accept

Thank you for making the improvements requested.